# Incentive Design for Multi-Agent Systems: A Bilevel Optimization Framework for Coordinating Independent Agents and Convergence Analysis

## Abstract

Incentive design aims to guide the performance of a system towards a human's intention or preference. We study this problem in a multi-agent system with one leader and multiple followers. Each follower independently solves a Markov decision process (MDP) to maximize its own expected total return with the same state space and action space. However, the leader's objective depends on the collective best-response policies of all followers. To influence these policies of followers, the leader provides side payments as incentives to individual followers at a cost, aiming to align the collective behaviors of followers with its own goal while minimizing this cost of incentive. Such a leader-followers interaction is formulated as a bilevel optimization problem: the lower level consists of followers individually optimizing their MDPs given the side payments, and the upper level involves the leader optimizing its objective function given the followers' best responses. The main challenge to solve the incentive design is that the leader's objective is generally non-concave and the lower level optimization problems can have multiple local optima. To this end, we employ a constrained optimization reformation of this bi-level optimization problem and develop an algorithm that provably converges to a stationary point of the original problem, by leveraging several smoothness properties of value functions in MDPs. We validate our algorithm in a stochastic gridworld by examining its convergence, verifying that the constraints are satisfied, and evaluating the improvement in the leader's performance.

## 1 Introduction

Incentive design aims to determine how an agent should incentivize a group of autonomous or semi-autonomous AI systems to adjust their behavior in alignment with human intentions, particularly when these systems have their own objectives, uncertainties, or interactions with other agents. The incentive design can be studied in the framework of the leader-follower games, also known as principal-agent games (Bolton & Dewatripont, 2005; Ho et al., 1981; Ho & Teneketzis, 1984; Simaan & Cruz Jr, 1973), where the follower encounters a planning problem that can be shaped by an incentive policy of the leader, and the leader is to design an incentive policy so that the follower's best response aligns with the leader's objective. This framework is widely used in mechanism design across various fields, such as economic markets (Myerson, 1981; Williams, 2011; Easley & Ghosh, 2015), online platforms (Ratliff et al., 2019), smart cities (Mei et al., 2017; Kazhamiakin et al., 2015), smart grids (Braithwait et al., 2006; Alquthami et al., 2021), and machine learning (Kang et al., 2023; Liu et al., 2024; Pásztor et al., 2024). For instance, Alquthami et al. (Alquthami et al., 2021) propose a mechanism for the fair design of customized price profiles for all users and tailored electricity tariffs for high-consumption customers. In the smart grid, to mitigate peak demand, the utility company may reduce prices at certain times to incentivize users, aiming to optimize social welfare while also improving profit. As another example, Liu et al. (Liu et al., 2024) formulate the data poisoning attack as a Stackelberg game, where the attacker (leader) crafts poisonous perturbations to the training data with the goal of reducing test accuracy, while the classifier (follower) optimizes its network parameters on the poisoned dataset.

In the real world, a leader may wish to steer a group of independent yet heterogeneous followers toward their intended preference. For example, in ride-sharing platforms, each passenger selects an optimal ride plan based on personal needs, while the platform adjusts incentives for certain plans to encourage behaviors such as frequent usage or early bookings. Motivated by such applications, we investigate incentive design in a hierarchical setting involving a single leader and multiple followers, where each follower's planning problem is modeled as a MDP. This raises the question: *how can the leader provide side payments to the group of followers to steer them closer to the intended preference while considering the cost of these incentives?* We address the problem through bilevel optimization. At the lower level, each follower independently selects an optimal policy with respect to a reward function shaped by side payments from the leader. At the upper level, the leader's objective depends on the collective best responses of the followers and involves balancing the benefit of influencing their behavior against the cost of providing incentives. Anticipating the followers' best responses, the leader aims to design an incentive policy that aligns the followers' behavior with his objective in a cost-effective manner.

## 1.1 Related work

Incentive design in MDPs has been investigated in the context of AI alignment, where the goal is to ensure that AI systems act consistently with human intentions. It is known that reward functions are often unintentionally and inevitably mis-specified, which can lead to harmful or undesirable behaviors such as reward hacking and goal misgeneralization (Ji et al., 2023). To achieve AI alignment, a reward or model parameter designer (the leader) guides the behavior of a learning agent (the follower) to improve system performance by aligning the agent's reward function with human knowledge or intrinsic rewards (Stadie et al., 2020). Chen et al. (Chen et al., 2022) study how to regulate an MDP agent when human wish to take the external costs/benefits of its actions into consideration. They formulate the problem as a bilevel program, where the upper-level model designer (leader) regulates the lower-level MDP (follower) by adjusting model parameters that influence the rewards and/or transition kernels. However, all of the aforementioned work assume a unique optimal solution in the lower-level problem and a single-leader-single-follower interaction. However, it is well-known that the optimal policy in an MDP may not be unique. Thus, we do not restrict the lower-level problem has a unique solution and consider how to design this incentive policy even if there are multiple optimal policies for the followers' MDPs.

For the incentive design problem in multi-agent system, Ratliff et. al. (Ratliff & Fiez, 2020) propose a method to adaptively design incentives for principal-agent problems in which the principal faces adverse selection in its interaction with multiple agents. They consider both the cases where agents play best response to one another (Nash) and where they employ myopic update rules. However, they study a different class of incentive design problems in which the follower's decision-making is modeled using a generalized linear model, as opposed to a MDP with a tunable reward function studied herein.

Since the incentive design could be cast as a bilevel optimization problem (Casorrán et al., 2019), we discuss recent work on bilevel optimization and the relation to our proposed method. In recent years (2022-), various studies have proposed algorithms to address bilevel optimization problems where the lower-level problem has multiple optimal solutions. For example, an algorithm was developed to converge to a stationary point of the bilevel problem using a first-order method (Kwon et al., 2023), and a primal-dual bilevel optimization (PDBO) approach demonstrated convergence to an optimal solution (Sow et al., 2022). However, these methods generally require assumptions about the convexity of objective functions, which may limit their applicability in broader contexts. Liu et al. (2022) develop a first-order algorithm for solving bi-level optimization problems with non-convex functions. In the MDP, the value function is generally non-concave under both direct and softmax parameterizations. However, given the L-smoothness of the value function and other mild conditions, we can show the assumptions required for the convergence results in Liu et al. (2022) are satisfied (section 3.3) and thus ensure the convergence of the proposed method for the class of incentive design problems.

### 1.2 Our contributions

- We study a class of incentive design problem in the presence of multiple followers where the followers may have multiple optimal strategies given the leader's incentive policy. Therefore, we can extend the application of the incentive design to more practical applications.

- To solve the Stackelberg equilibrium, standard learning dynamics require computing the total derivative of the leader's objective function via the implicit function theorem (Fiez et al., 2020). This total derivative involves computing the inverse of the Hessian matrix, which is computationally expensive and often impractical to evaluate at each iteration. To tackle this challenge, we employ the first-order method in Liu et al. (2022) and reformulate the original problem as a constrained optimization problem. We then develop a first-order algorithm for computing a stationary point for both direct parameterization and softmax parameterization of the policy, and derive the corresponding gradient computation, which can be obtained directly or estimated using standard RL techniques.

- We leverage smoothness properties of the value functions in MDP to prove the convergence of this first order method, provided the followers' best responses are restricted to two classes of policy spaces (direct parameterization and softmax parameterization).

- We demonstrate the algorithm's effectiveness by showing the convergence of the algorithm and satisfaction of the constrains, and the improvement in the leader's performance in a stochastic gridworld environment using two different policy parameterization methods.

## 2 Preliminaries and problem formulation

**Notation** Throughout the article, we adopt the following notations. $\|\cdot\|$ refers to the $L^2$ norm in this paper. The space of probability distributions on the set $S$ is denoted by $\mathcal{D}(S)$. We use superscripts to indicate time steps. For instance, a history of state-action pairs is represented as $((s^{(0)}, a^{(0)}), \cdots, (s^{(n)}, a^{(n)}))$. We use the subscript to indicate the variable associated with the follower, and boldface notation to denote joint state, action, and policy. For instance, the joint state $\mathbf{s}$ denotes $(s_1, \cdots, s_n)$.

### 2.1 Preliminaries

A single-agent MDP is defined as a tuple $M = (S, A, P, \mu, \gamma, \bar{R})$, where $S$ is a finite set of states, $A$ is a finite set of actions, $P \colon S \times A \to \mathcal{D}(S)$ is a probabilistic transition function such that $P(s'|s, a)$ is the probability of reaching state $s'$ given action $a$ being taken at state $s$, $\mu \in \mathcal{D}(S)$ is the initial distribution, $\gamma \in [0, 1]$ is the discount factor, and $\bar{R} \colon S \times A \to \mathbf{R}$ is the original reward function such that $\bar{R}(s, a)$ is the reward received by the follower for taking action $a$ in state $s$.

Let $\pi : S \times A \to [0, 1]$ be a stochastic, Markovian policy that specifies, for each state $s \in S$, a probability distribution over the actions $a \in A$. The value function of a Markov policy $\pi$ given reward $R : S \times A \to \mathbf{R}$ is defined by

$$V(\mu, \pi) = \mathbb{E}_\pi \left[ \sum_{k=0}^\infty \gamma^k R(S^{(k)}, A^{(k)}) | S^{(0)} \sim \mu \right].$$

The Q-value function given policy $\pi$ $Q \colon S \times A \to \mathbf{R}$ is defined as:

$$Q(s, a, \pi) = \mathbb{E}_\pi \left[ \sum_{k=0}^\infty \gamma^k R(S^{(k)}, A^{(k)}) | S^{(0)} = s, A^{(0)} = a \right].$$

The state-action visitation distribution $d_\mu^\pi(s, a)$ of a policy $\pi$ is defined as:

$$d_\mu^\pi(s, a) = (1 - \gamma) \sum_{k=0}^\infty \gamma^k \mathrm{Pr}^\pi(S^{(k)} = s, A^{(k)} = a | S^{(0)} \sim \mu),$$

where $\mathrm{Pr}^\pi(S^{(k)} = s, A^{(k)} = a | S^{(0)} \sim \mu)$ is the probability of visiting state $s$ and taking action $a$ at the $k$-th time step when the agents follow policy $\pi$ with an initial state distribution $\mu$.

**The followers' model**  We model the behavior of $n$ distinct followers using $n$ different MDPs. While these MDPs share the same state space and action space, they differ in their transition probabilities, reward functions, initial distributions, and discounted factors. Let $i \in \mathbf{N} = \{1, \ldots, n\}$ be the index of a follower, then the MDP for follower $i$ is denoted as

$$M_i = (S, A, P_i, \mu_i, \gamma_i, \bar{R}_i), \forall i \in \mathbf{N}.$$

**Assumption 1.** *The follower $i$ solves his optimal policy in its MDP $M_i$ independently of follower $j$, for any $i, j \in \mathbf{N}$. One follower's action does not affect the state of other followers. Additionally, all followers arrive at states and take actions simultaneously, starting at $t = 0$.*

**The leader's model**  By Assumption 1, all followers are independent in their transition dynamics and reward functions. However, their collective behaviors affect the leader's value in the following way. The leader's decision problem can be viewed from a multi-agent MDP:

$$M = (\mathbf{S}, \mathbf{A}, \mathbf{P}, \boldsymbol{\mu}, \gamma, R_l),$$

where $\mathbf{S} \triangleq S^n$ denote the joint state space (each $s^n \in \mathbf{S}$ is the followers' collection of states), $\mathbf{A} \triangleq A^n$ denote the joint action space (each $a^n \in \mathbf{A}$ is the followers' collection of actions, $P \colon \mathbf{S} \times \mathbf{A} \to \mathcal{D}(\mathbf{S})$ is a probabilistic joint transition function such that $\mathbf{P}(\mathbf{s}'|\mathbf{s}, \mathbf{a}) = \prod_{i \in \mathbf{N}} P_i(s'|s, a)$ which is the probability of reaching followers' joint state $\mathbf{s}'$ given followers' joint action $\mathbf{a}$ being taken at followers' joint state $\mathbf{s}$, $\boldsymbol{\mu}$ is the followers' joint initial distribution such that $\boldsymbol{\mu}(\mathbf{s}) = \prod_{i \in \mathbf{N}} \mu_i(s_i)$, $\gamma \in [0, 1]$ is the discount factor, and $R_l \colon \mathbf{S} \times \mathbf{A} \to \mathbf{R}$ is the reward function such that $R_l(\mathbf{s}, \mathbf{a})$ is the reward received by the leader for the $n$ followers taking joint action $\mathbf{a}$ at joint state $\mathbf{s}$.

Given the followers' the joint policy $\boldsymbol{\pi} \colon \mathbf{S} \to \mathcal{D}(\mathbf{A})$ such that $\boldsymbol{\pi}(\mathbf{a}|\mathbf{s}) = \prod_{i \in \mathbf{N}} \pi_i(a_i|s_i)$, the leader's value function is then defined as

$$V(\boldsymbol{\mu}, \boldsymbol{\pi}) = \mathbb{E}_{\boldsymbol{\pi}} \left[ \sum_{k=0}^{\infty} \gamma^k R_l(\mathbf{S}^{(k)}, \mathbf{A}^{(k)}) \mid \mathbf{S}^{(0)} \sim \boldsymbol{\mu} \right]. \tag{1}$$

**Incentives as side payments**  The leader can incentivize the followers to align their behavior with the leader's objective. Leader's tailored incentive to the group of followers is represented as a function $x \colon \mathbf{N} \times S \times A \to \mathbf{R}_+$, hereafter referred to as the *side payments*. Specifically, $x(i, s, a)$ is the additional reward that leader offers to the follower $i$ when follwer $i$ takes action $a$ in state $s$. We can view $x$ as a vector with the entry to be $x(i, s, a)$ for each $i \in \mathbf{N}$, $s \in S$, $a \in A$. We denote the set of vector $x$ as $\mathcal{X}$. The side payments for the follower $i$ are denoted as $x_i \colon S \times A \to \mathbf{R}_+$, where $x_i(s, a) = x(i, s, a)$. The vector $x_i$ can also be viewed as a vector in a similar way to $x$.

Given a side payments $x_i$, the follower $i$'s modified reward function $R_i(x_i)$ is defined as follows. For all $(s, a) \in S \times A$,

$$R_i(s, a; x_i) = \bar{R}_i(s, a) + x_i(s, a). \tag{2}$$

As a result, the follower $i$'s planning problem with side payments $x_i$ is an MDP with modified reward:

$$M_i(x_i) = (S, A, P_i, \mu_i, \gamma_i, R_i(x_i)).$$

Given the follower $i$'s policy $\pi_i \colon S \to \mathcal{D}(A)$, his value function is defined as:

$$V_i(\mu_i, \pi_i; x_i) = \mathbb{E}_{\pi_i} \left[ \sum_{k=0}^{\infty} \gamma_i^k R_i(S_i^{(k)}, A_i^{(k)}; x_i) | S_i^{(0)} \sim \mu_i \right]. \tag{3}$$

## 2.2  Problem statement

**Policy parameterization**  We now introduce two common parametric policy classes $\{\pi_\theta | \theta \in \Theta\}$ which are complete in the sense that any Markov policy can be represented in each class. The two classes are as follows:

- Direct parameterization: The polices are parameterized by:

$$\pi_\theta(a|s) = \theta_{s,a},$$

where $\theta_s \in \mathcal{D}(A)$, for all $s \in S$.

- Softmax parameterization: For unconstrained $\theta_{s,a} \in \mathbf{R}$,

$$\pi_\theta(a|s) = \frac{\exp(\theta_{s,a}/\tau)}{\sum_{a' \in A} \exp(\theta_{s,a'}/\tau)},$$

where $\tau$ is the temperature parameter that determines how closely the softmax function approximates the hardmax function.

**Problem 1.** *Assuming follower $i$'s policy is parameterized by vector $\theta_i \in \Theta_i$, we denote vector $\theta \triangleq [\theta_1, \cdots, \theta_n] \in \Theta$. The incentive design problem is formulated as the following bilevel optimization problem:*

$$\max_{x \in \mathcal{X}, \theta \in \Theta} \quad V(\boldsymbol{\mu}, \boldsymbol{\pi}_\theta) - w \cdot C(x) \tag{4}$$
$$\text{s.t. } \theta_i \in \arg\max_{\theta_i \in \Theta_i} V_i(\mu_i, \pi_{\theta_i}; x_i), \forall i \in \mathbf{N},$$

*where $w$ is the regularization factor, and $C: \mathcal{X} \to \mathbf{R}_+$ is a cost function for side payments.*

In the following, when the initial state distributions are clear from the context, we omit them: $V(\boldsymbol{\pi}_\theta) \triangleq V(\boldsymbol{\mu}, \boldsymbol{\pi}_\theta)$, $V_i(\pi_{\theta_i}; x_i) \triangleq V_i(\mu_i, \pi_{\theta_i}; x_i), \forall i \in \mathbf{N}$.

## 3 A bilevel optimization approach and convergence analysis

### 3.1 A reformulation to a constrained optimization problem

To maximize the leader's objective function with his decision variable $x$, we reformulate the bilevel optimization problem 4 to a constrained optimization problem.

First, due to the independent dynamics and decision-making processes of the followers, it is easy to show that problem 4 is equivalent to

$$\max_{x \in \mathcal{X}, \theta \in \Theta} \quad V(\boldsymbol{\pi}_\theta) - w \cdot C(x) \tag{5}$$
$$\text{s.t. } \theta \in \arg\max_{\theta' \in \Theta} \sum_{i \in \mathbf{N}} V_i(\pi_{\theta'_i}; x_i).$$

Let $V_i^*(x_i) \triangleq \max_{\theta_i \in \Pi_i} V_i(\pi_{\theta_i}; x_i)$, then problem 5 is equivalent to:

$$\max_{x \in \mathcal{X}, \theta \in \Theta} \quad V(\boldsymbol{\pi}_\theta) - w \cdot C(x) \tag{6}$$
$$\text{s.t. } \sum_{i \in \mathbf{N}} V_i(\pi_{\theta_i}; x_i) - \sum_{i \in \mathbf{N}} V_i^*(x_i) \geq 0.$$

The constraint forces $\theta$ to be within the set that maximizes the value achieved by the followers. It is noted that $\sum_{i \in \mathbf{N}} V_i(\pi_{\theta'_i}; x_i) - \sum_{i \in \mathbf{N}} V_i^*(x_i) = 0$ for any $\theta'_i \in \arg\max_{\theta_i} V_i(\pi_{\theta'_i}; x_i)$, and $\sum_{i \in \mathbf{N}} V_i(\pi_{\theta'_i}; x_i) - \sum_{i \in \mathbf{N}} V_i^*(x_i) < 0$, for any $\theta'_i \notin \arg\max_{\theta_i} V_i(\pi_{\theta'_i}; x_i)$. A feasible solution $(x, \theta)$ to problem 6 ensures that $\theta$ is the collection of best responses of followers with the leader's decision variable $x$.

Intuitively, the leader is to determine jointly the side payments $x$ and the policy parameters $\theta$ in the constraint sets. When choosing a follower's policy, the leader must ensure the chosen policy performs as good as the follower's best response to the side payments $x$, for each individual follower, to satisfy the constraint.

**Remark 1.** *We can show that the solution to problem 6 is a Strong Stackelberg equilibrium. In a Stackelberg game, the leader chooses an action $u \in U$, and then the follower, after observing $u$, selects an action $v \in V$. The follower's payoff is denoted by $h(u, v)$, and the set of the follower's best responses to a given leader action $u$ is*

$$S(u) = \arg\max_{v \in V} h(u, v).$$

*The leader's payoff is denoted by $H(u, v)$, and the leader's objective is to maximize this payoff, taking into account the follower's best response.*

*In the Strong Stackelberg Equilibrium (SSE), the follower is assumed to break ties in favor of the leader (i.e. select the best response that is most favorable to the leader). The set of best responses in optimistic position is defined as:*

$$S^o(u) = \arg\max_{v \in V} \{H(u, v) : v \in S(u)\}.$$

*Then the bilevel optimization problem to solve SSE is defined as:*

$$\max_{u \in U, v \in V} \quad H(u, v)$$
$$\text{s.t.} \quad v \in S^o(u)$$

*In the Weak Stackelberg Equilibrium (WSE), the follower is assumed to break ties against the leader, and the leader will have to prepare for the worst. The set of follower's best responses in pessimistic position is defined as:*

$$S^p(u) = \arg\min_{v \in V} \{H(u, v) : v \in S(u)\}.$$

*Then the bilevel optimization problem to solve WSE is defined as:*

$$\max_{u \in U, v \in V} \quad H(u, v)$$
$$\text{s.t.} \quad v \in S^p(u).$$

*Since both the incentive variable and the follower's policy parameter maximize the leader's objective function, the solution to the reduced constrained optimization problem 6 is to solve the SSE. This aligns with the setting in which followers are willing to adopt policies that benefit the leader's objective.*

## 3.2 The algorithm and gradient computation

To notational convenience, let the objective function be denoted by

$$f(x, \theta) \triangleq V(\boldsymbol{\pi}_\theta) - w \cdot C(x),$$

and the constraint function be denoted by

$$q(x, \theta) \triangleq g(x, \theta) - g^*(x),$$

where $g(x, \theta) \triangleq \sum_{i \in \mathbf{N}} V_i(\pi_i; x_i)$ and $g^*(x) \triangleq \sum_{i \in \mathbf{N}} V_i^*(x_i)$.

We adopt a bilevel optimization algorithm (Liu et al., 2022) to solve the constrained optimization problem 6 described in the Algorithm 1.

To implement this Algorithm, we first compute the gradient (or partial derivative) of the objective function $f$ and constraint function $q$ assuming $C(x)$ are differentiable.

The gradient of $f(x, \theta)$ w.r.t. $\theta$ is equivalent to the policy gradient of leader's value function, *i.e.*,

$$\nabla_\theta f(x, \theta) = \nabla_\theta V(\boldsymbol{\pi}_\theta) = \frac{1}{1 - \gamma} \mathbb{E}_{\mathbf{s} \sim d_\mu^{\boldsymbol{\pi}_\theta}} \mathbb{E}_{\mathbf{a} \sim \boldsymbol{\pi}_\theta} [\nabla_\theta \log \boldsymbol{\pi}_\theta(\mathbf{a}|\mathbf{s}) Q(\mathbf{s}, \mathbf{a}, \boldsymbol{\pi}_\theta)]. \tag{7}$$

---

**Algorithm 1** Bilevel Optimization Algorithm

---

**Require:** Step size $\xi$, positive constant $\eta$, known variables in followers' MDPs, initialization $(x^{(0)}, \theta^{(0)}, \lambda^{(0)})$.

1: **for** iteration $t = 0, 1, 2, \ldots$ **do**

2:     Update $\lambda^{(t+1)}$ according to

$$\lambda^{(k)} = \max\left(\eta - \frac{\langle \nabla f(x^{(k)}, \theta^{(k)}), \nabla q(x^{(k)}, \theta^{(k)}) \rangle}{\|\nabla q(x^{(k)}, \theta^{(k)})\|^2}, 0\right).$$

3:     Update $x^{(t+1)}$ according to

$$x^{(t+1)} = \mathsf{Proj}_{\mathcal{X}}[x^{(k)} + \xi(\nabla_x f(x^{(k)}, \theta^{(k)}) + \lambda^{(k)} \nabla_x q(x^{(k)}, \theta^{(k)}))].$$

4:     Update $\theta^{(t+1)}$ according to

$$\theta^{(t+1)} = \mathsf{Proj}_{\Theta}[\theta^{(k)} + \xi(\nabla_\theta f(x^{(k)}, \theta^{(k)}) + \lambda^{(k)} \nabla_\theta q(x^{(k)}, \theta^{(k)}))].$$

5: **end for**

6: **return** $x_T$, $\theta_T$.

---

The policy gradient update is based on the REINFORCE estimator:

$$\hat{\nabla}_\theta V(\boldsymbol{\pi}_\theta) = \frac{1}{m} \sum_{j=1}^{m} \sum_{k=1}^{|\tau_j|} \nabla_\theta \log \boldsymbol{\pi}_\theta(\mathbf{a}^{(k)} \mid \mathbf{s}^{(k)}) R(\tau_j), \tag{8}$$

where $R\left(\tau^{(i)}\right)$ is the total return with the sampled trajectory $\tau_j$, and $|\tau_j|$ is the length of sampled trajectory $\tau_j$.

Under direct parameterization,

$$\frac{\partial \log \boldsymbol{\pi}_\theta(\mathbf{a}|\mathbf{s})}{\partial \theta_{i,s,a}} = \frac{\partial \log \pi_{\theta_i}(a_i|s_i)}{\partial \theta_{i,s,a}} = \begin{cases} 1/\theta_{i,s,a}, & \text{if } (s_i, a_i) = (s, a) \\ 0, & \text{otherwise} \end{cases} \tag{9}$$

Under softmax parameterization,

$$\frac{\partial \log \boldsymbol{\pi}_\theta(\mathbf{a}|\mathbf{s})}{\partial \theta_{i,s,a}} = \frac{\partial \log \pi_{\theta_i}(a_i|s_i)}{\partial \theta_{i,s,a}} = \begin{cases} \frac{1}{\tau}(1 - \pi_{\theta_i}(s, a)), & \text{if } (s_i, a_i) = (s, a) \\ -\frac{1}{\tau}\pi_{\theta_i}(s, a), & \text{if } s_i = s, \ a_i \neq a \\ 0, & \text{otherwise} \end{cases} \tag{10}$$

The gradient of the objective function w.r.t. $x$ is simply

$$\nabla_x f(x, \theta) = -w \cdot \nabla_x C(x), \tag{11}$$

To derive $\nabla_x q(x, \theta)$, we first provide the following lemma:

**Lemma 1.** *Consider a reward function with side payments $R(s, a; x) = \bar{R}(s, a) + x(s, a)$ where $\bar{R}$ is the original reward in MDP $M$, the partial derivative of a value function $V(\mu, \pi_\theta; x)$ w.r.t. the side payments $x(s, a)$ is state-action visitation $d_\mu^\pi(s, a)$. That is*

$$\frac{\partial V(\mu, \pi_\theta; x)}{\partial x(s, a)} = d_\mu^{\pi_\theta}(s, a).$$

The proof is given in Appendix A.

Given the policy $\pi$ and initial distribution $\mu$, we use $\mathbf{d}_\mu^\pi$ to denote the vector with the entries to be $d_\mu^\pi(s, a)$ for each $s \in S$ and $a \in A$. Then by Lemma 1, the gradient of $q(x, \theta)$ can be denoted as:

$$\nabla_x q(x, \theta) = [\mathbf{d}_{\mu_1}^{\pi_{\theta_1}} - \mathbf{d}_{\mu_1}^{\pi_{\theta_1^*(x_1)}}, \ldots, \mathbf{d}_{\mu_n}^{\pi_{\theta_n}} - \mathbf{d}_{\mu_n}^{\pi_{\theta_n^*(x_n)}}], \tag{12}$$

where $\theta_i^*(x_i) \in \arg\max_{\theta_i} V_i(\pi_{\theta_i}; x_i)$, $i \in \mathbf{N}$.

Because of the independence of the followers (i.e., $\theta_i$ only plays a role in $V_i$) and the fact that $g^*(x)$ is independent of $\theta$, we can express the partial derivative of $q(x, \theta)$ w.r.t. each policy parameter $\theta_{i,s,a}$ as follows:

$$\frac{\partial q(x, \theta)}{\partial \theta_{i,s,a}} = \frac{\partial g(x, \theta)}{\partial \theta_{i,s,a}} = \frac{\sum_{i \in \mathbf{N}} V_i(\pi_{\theta_i}; x_i)}{\partial \theta_{i,s,a}} = \frac{\partial V_i(\pi_{\theta_i}; x_i)}{\partial \theta_{i,s,a}}, \tag{13}$$

which can be computed using the policy gradient method in the follower $i$'s MDP.

### 3.3 Convergence analysis

It is worth noting that the problem 6 can have multiple stationary points. We use a simple example to illustrate this.

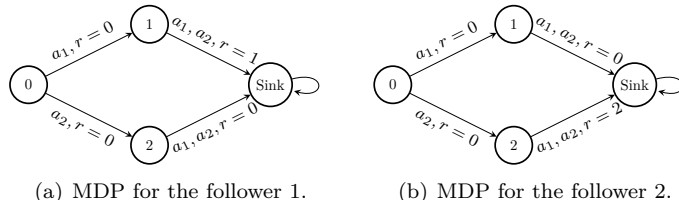

(a) MDP for the follower 1.    (b) MDP for the follower 2.

Figure 1: A small MDP example showing two followers with different reward functions.

**Example 1.** *Figure 1 illustrates two followers' deterministic dynamics and reward functions in a small MDP with infinite horizen. The state space $S$ is $\{0, 1, 2, Sink\}$, and action space is $\{a_1, a_2\}$. Both the leader and the followers receive a reward of 0 in the "Sink" state. Follower 1 and 2 has state 1 and state 2 as their respective goal states in the individual MDPs. We set the leader's reward function to be state-dependent: The leader receives a reward of 6 when joint state is $(2, 1)$. A reward of 5 is obtained when the joint state is either $(2, 0)$ or $(0, 1)$. For all other joint states, the leader's reward is 0. The leader's reward can be understood as follows: The leader receives a reward of 6 if both followers reach each other's goal states, 5 if only one does, and 0 if neither does.*

*We assume that the leader can give non-negative side payments to follower 1 at state 2, and follower 2 at state 1. We also assume that the discounting factor in leader's value is 1. We denote the side payments as $x = (x_1, x_2)$. The leader's cost function is chosen to be the summation of side payments.*

*In this case, for any fixed side payments $x_1 \neq 1$ and $x_2 \neq 2$, each follower has a unique best response. In other cases, although a follower may have multiple optimal solutions, the leader selects the one that maximizes its objective function. Consequently, the incentive design problem in this example reduces to a single-level optimization problem, which aims to maximize $f_m(x) \triangleq \max_\theta f(x, \theta)$ w.r.t. $x$, where $f(x, \theta)$ denotes the leader's objective function.*

*Under direct parameterization, $f_m(x)$ can be expressed as the following piece-wise linear function:*

$$f_m(x) = \begin{cases} -x_1 - x_2, & \text{if } 0 \leq x_1 < 1, \ 0 \leq x_2 < 2 \\ 5 - x_1 - x_2, & \text{if } 0 \leq x_1 < 1, \ x_2 \geq 2, \text{ or } x_1 \geq 1, \ 0 \leq x_2 < 2 \\ 6 - x_1 - x_2, & \text{if } x_1 \geq 1, \ x_2 \geq 2 \end{cases}$$

*The stationary points of this function are $(0,0)$, $(1,0)$, $(0,2)$, and $(1,2)$, as illustrated in the contour plot shown in Figure 2(a).*

*Under softmax parameterization, choose $\theta^*_{s,a} = Q^*(s,a,x)$. We denote $\pi_1(a_1|0) = p_1$ for follower 1, and $\pi_2(a_1|0) = p_1$ for follower 2. Let $\tau = 0.05$, $\gamma_1 = \gamma_2 = 1$, we have the following relation between the side payments $x$ and the probability $p_1, p_2$:*

$$p_1(x) = \frac{\exp(20)}{\exp(20) + \exp(20x_1)}, \quad p_2(x) = \frac{\exp(20x_2)}{\exp(20x_2) + \exp(40)}.$$

*Then, $f_m(x) = 5 + 4p_1(x)p_2(x) - 5p_1(x) + p_2(x) - (x_1 + x_2)$. Figure 2(b) illustrates the function's contour plot, revealing multiple stationary points across its domain.*

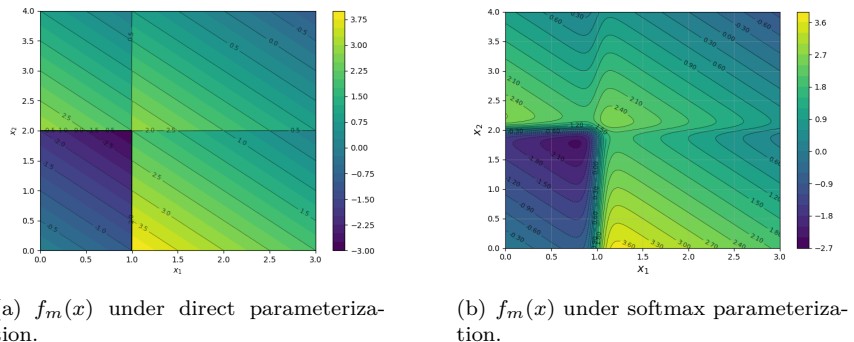

(a) $f_m(x)$ under direct parameterization.

(b) $f_m(x)$ under softmax parameterization.

Figure 2: Contour plots of $f_m(x)$ under direct and softmax parameterizations.

We aim to prove that the limit of the sequence $(\pi^{(k)}, x^{(k)})_{k=0}^{\infty}$ generated from the Algorithm 1 is a stationary point of the bilevel optimization problem (equation 6) for both direct and softmax parameterization. Given that the value function is non-concave, as shown in Lemma 1 of Agarwal et al. (2021), our analysis focus on examining the smoothness properties of both the objective and constraint functions in equation 6.

To establish the proof, we begin by stating the following assumptions.

**Assumption 2.** *The cost function $C \colon \mathcal{X} \to \mathbf{R}_+$ is differentiable, and satisfies two conditions:*

- *$C$ and $\nabla C$ is Lipschitz continuous.*
- *$|C(x)|$ is upper bounded by a constant $\beta < \infty$.*

**Assumption 3.** *We assume that the reward functions of all the followers and the leader are bounded, and the norm of side payments $||x||$ is bounded, and as a result, given $\pi$, the leader's value function $V(\boldsymbol{\pi}_\theta)$, as well as each follower's value function $V(\pi_\theta; x)$ for each follower are all bounded.*

Recall that $f(x,\theta) \triangleq V(\boldsymbol{\pi}_\theta) - C(x)$, $g(x,\theta) \triangleq \sum_{i \in \mathbf{N}} V_i(\pi_i; x_i)$, $g^*(x) \triangleq \sum_{i \in \mathbf{N}} V_i^*(x_i)$, and $q(x,\theta) \triangleq g(x,\theta) - g^*(x)$.

We now derive three key corollaries in preparation for the final result.

**Corollary 1.** *There exists a constant $0 < \beta < \infty$ such that $\|\nabla f(x,\theta)\|$, $\|\nabla g(x,\theta)\|$, $|f(x,\theta)|$ and $|g(x,\theta)|$ are all upper bounded by $\beta$ for any $(x,\pi)$.*

*Proof.* The gradient computations are detailed in Section 3.2. The proof then follows straightforwardly from Assumption 2 and Assumption 3 and is therefore omitted. □

**Corollary 2.** *$\nabla f$ and $\nabla g$ are Lipschitz continuous w.r.t. the joint inputs $(x,\theta)$.*

*Proof.* See the proof in Appendix C. □

**Corollary 3.** *Given a set of followers' optimal policies $\{\pi_{\theta_i^*}\}_{i=1}^n$, $\nabla_\theta g(x,\theta)$ satisfies the PL-inequality w.r.t. $\theta$. i.e. there exists $\kappa > 0$ such that for any $(x,\theta)$,*

$$\|\nabla_\theta g(x,\theta)\|^2 \geq \kappa(g(x,\theta^*) - g(x,\theta)).$$

*Proof.* See the proof in Appendix F. ☐

The measure of stationary, introduced in Liu et al. (2022), is adapted to our setting as follows.

**Definition 1.** *The measure of stationary is defined as*

$$\mathcal{K}(x,\theta) = \min_{\lambda \geq 0} \|\nabla f(x,\theta) + \lambda \nabla q(x,\theta)\| - q(x,\theta).$$

The following theorem, paraphrases Theorem 2 in (Liu et al., 2022) guarantees that the algorithm generates a sequence $(\theta^{(k)}, x^{(k)})_{k=0}^\infty$ that satisfies $\mathcal{K}(\theta^{(k)}, x^{(k)}) \to 0$ as $k \to 0$ for the bilevel optimization problem 6,.

**Theorem 1.** *Consider Algorithm 1 with $\xi \leq 1/L$ (where L is the L-Lipschitz constant defined in Corollary 2). With Assumption 2, Assumption 3, and that $q(x,\theta)$ is differentiable on $(x,\theta)$ at every iteration $k \geq 0$. Then there exists a constant c depending on $\xi$, $\kappa$ (where $\kappa$ is the PL inequality constant defined in Corollary 3), $\eta$, L, such that, we have for any iteration numbers $K \geq 0$*

$$\min_{k \leq K} \mathcal{K}(\theta^{(k)}, x^{(k)}) = O\left(\sqrt{\xi} + \sqrt{\frac{1}{\xi K}}\right),$$

*where b is a positive constant depending on $\kappa$, L, and $\xi$.*

*Proof.* With Corollary 1, Corollary 2, and Corollary 3, the optimization problem satisfies the condition for convergence of a constrained optimization reformation for bi-level optimization problems (Liu et al., 2022). ☐

## 4 Experiment results

We consider a multi-agent stochastic gridworld shown in Figure 3. The robots can move in four compass directions. Given an action, say, "N", robot 1 (resp. robot 2) enters its intended cell with a probability of $1-2\alpha_1$ (resp. $1-2\alpha_2$), while entering the neighboring cells (west and east) with probabilities $\alpha_1$ (resp. $\alpha_2$). The original reward structure for the robots is state-based: they receive $-5$ at fire states, 8 at less desired goal states, and 10 at the goal state. The two robots with different dynamics aims to maximize the total reward. After the robot reaches either the goal state or a less desired goal state, it reaches a state 'Sink' with probability 1 and its interaction with the gridworld is terminated.

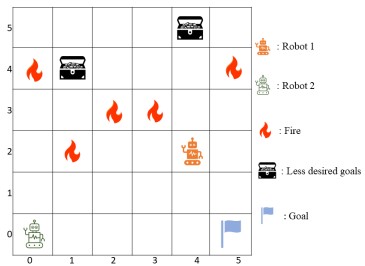

Figure 3: A $6 \times 6$ gridworld.

The leader's reward $R_l(\mathbf{s}, \mathbf{a})$ depends on the joint state $\mathbf{s} = (s_1, s_2)$. Let $\mathcal{F} = \{(1,4), (4,5)\}$ denotes the set of less desired goals for the followers.
The leader receives a reward of 10 when both $s_1$ and $s_2$ lie in $\mathcal{F}$. If one follower reaches a state in set $\mathcal{F}$ while the other is at the "Sink" state, the leader receives a reward of 8. When one follower reaches a state in set $\mathcal{F}$ and the other is not at the "Sink" state, the reward is 2. In all remaining cases, the leader obtains a reward of 0. Based on this reward structure, if we set the discounted factor in leader's value to be 1, the leader will receive 10 if both robots end up in less desired goal states, 8 if a robot reaches a less desired goal state after another robot reaches the goal state, and 2 if before. With this leader's reward function, the leader wants to have two robots visit the less desired goals instead of the flag goal state together. To achieve this outcome, the leader is to incentivize the robots with side payments at some costs. The side payment to robot 1 is designed to be placed at $(4,5)$, and side payment to robot 2 is designed to be placed at $(1,4)$.

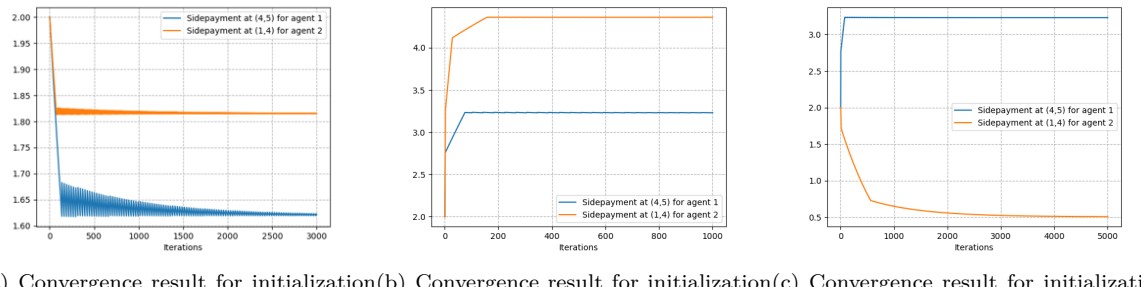

(a) Convergence result for initialization 1.     (b) Convergence result for initialization 2.     (c) Convergence result for initialization 3.

Figure 4: Experiment environment and the convergence plots for different initializations under direct parameterization.

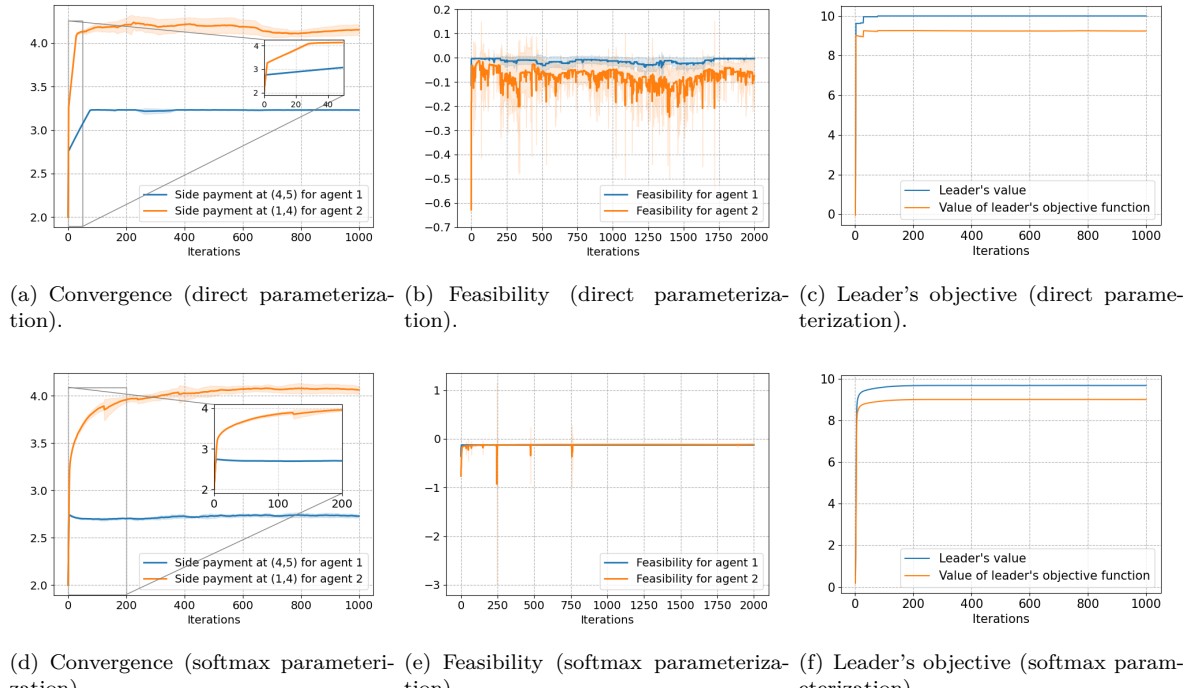

(a) Convergence (direct parameterization).

(b) Feasibility (direct parameterization).

(c) Leader's objective (direct parameterization).

(d) Convergence (softmax parameterization).

(e) Feasibility (softmax parameterization).

(f) Leader's objective (softmax parameterization).

Figure 5: Trends in convergence, feasibility, and leader's objective across replicated experiments with initialization (2) under direct and softmax parameterizations.

We choose the cost function in leader's objective function to be $C(x) = \sum_{i,s,a} x_{i,s,a}$, and $w = 0.1$ in equation 4 for balancing maximizing leader's reward and minimizing the cost of incentive. Then, we apply Algorithm 1 to compute a stationary point to equation 4 and determine the leader's incentive design $x$. The experiments validate the convergence of the side payments vector, and the performance improvement of the leader.

Table 1 summarizes the experiments results under direct parameterization starting with three different initializations (all values are reported after convergence): (1) The initial policy are set to the followers' best response policies given initial side payments. (2) The initial policies are designed to maximize the leader's value; and (3) The initial policy for follower 2 is designed to have a low probability of reaching the less desired goal states, while the initial policy for follower 1 is configured to ensure a high probability of reaching the less desired goal states. All the experiments begin with initial side payments of 2 for both robots. Figure 4 shows the convergence trends of the side payments for the three different initializations. Because the problem has

Table 1: Summary of experiment results when under direct parameterization.

| Initialization | $x_1$ | $x_2$ | Feasibility | Leader's value | Leader's objective |
|---|---|---|---|---|---|
| 1 | 1.6217 | 1.8154 | -1.4121e-06 | 0.4145 | 0.0708 |
| 2 | 3.2322 | 4.3604 | -2.6256e-04 | 9.6954 | 8.9361 |
| 3 | 3.2299 | 0.5067 | -1.3950e-04 | 2.1214 | 1.7477 |

[a] Feasibility is the value of constraint function in the incentive design problem.
[b] Leader's value is the value of leader's value function (without the cost of side payments).
[b] Leader's objective is the value of leader's objective function (with the cost of side payments).

multiple stationary points, the algorithm may converge to different points depending on its initialization. Particularly, we present more detailed results for experiment (2) under direct and softmax parameterization in Figure 5. Figure 5(a), 5(d) show the trends of convergence, Figure 5(b), 5(e) validates that the solutions are feasible and satisfy the constraint, and Figure 5(c), 5(f) evaluate the leader's value and the value objective function for both policy parameterizations, showing a significant performance improvement of leader. All the results show that the algorithm converges to a stationary point, and the final policies satisfy the constraints while increasing the value of the leader's objective function.

Figure 6 presents the occupancy-measure heatmaps for each robot's optimal policy with side payments at both the first and final iterations, for the experiment under initialization 2 in Table 1. The results show that our algorithm significantly shifts the followers' behaviors toward the leader's preference through the use of side payments.

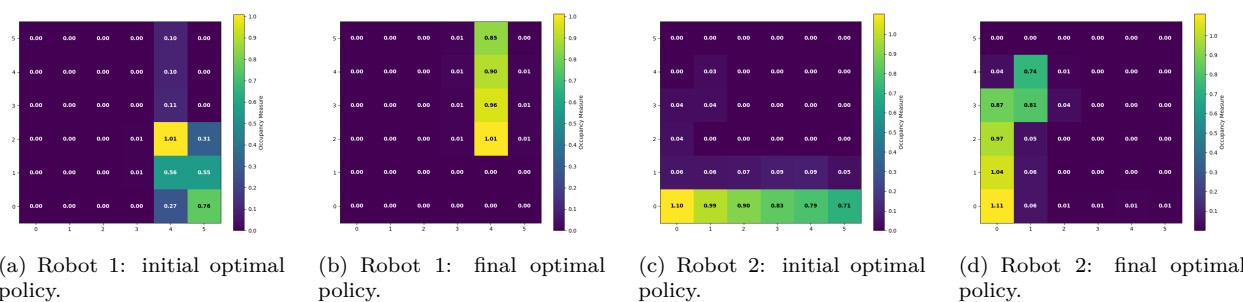

(a) Robot 1: initial optimal policy.

(b) Robot 1: final optimal policy.

(c) Robot 2: initial optimal policy.

(d) Robot 2: final optimal policy.

Figure 6: Comparison of occupancy measures between the first iteration and last iteration for two robots.

## 5 Conclusion

This paper studies incentive design for multiple independent followers whose aggregated behaviors determine the leader's value. We allow for flexibility in the agents' policies in the lower level problem, assuming that each agent may have multiple optimal policies. We first formulate the problem as a bilevel optimization problem to find the Strong Stackelberg Equilibrium. Solving this bilevel optimization problem presents challenges due to non-convexity in both the objective function and constraint functions. To address this, we transform the original bilevel problem into a constraint optimization problem and propose an algorithm inspired by existing bi-level optimization solutions. Our main contribution is to prove that the algorithm converges to a stationary point of the original incentive design problem, leveraging key properties of policy gradients and value functions in single-agent MDPs. The experimental results demonstrate that the algorithm converges, the constraints are satisfied, and the leader's performance improves, thereby validating our theoretical convergence proof under both direct and softmax parameterizations.

Future directions include solving the Weak Stackelberg Equilibrium for this class of leader–multi-follower games, where followers may not act cooperatively with the leader, leading to a min–max formulation for the leader's problem. Another direction is to consider adaptive incentive design where the leader has partial or incomplete information about the reward functions or transition functions in the followers' Markov decision processes.

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

## A  Proof of Lemma 1

*Proof.* The value function can be expressed using the state-action visitation distribution and reward (Altman, 2021):

$$V(\mu, \pi_\theta; x) = \sum_{s \in S, a \in A} d_\mu^\pi(s, a) R(s, a; x).$$

As we recall that $R(s, a; x) = \bar{R}(s, a) + x(s, a)$, we have

$$\frac{\partial V(\mu, \pi_\theta; x)}{\partial x_{\tilde{s}, \tilde{a}}} = \frac{\partial}{\partial x_{\tilde{s}, \tilde{a}}} \left( \sum_{(s,a) \in S \times A} d_\mu^\pi(s, a) R(s, a; x) \right) = d_\mu^\pi(\tilde{s}, \tilde{a}) \nabla_{x(\tilde{s}, \tilde{a})} R(\tilde{s}, \tilde{a}; x) = d_\mu^\pi(\tilde{s}, \tilde{a}).$$

□

## B  State visitation distributions, and the gradient and smoothness property of value function in a single-agent MDP

To lay the groundwork for the proofs, we introduce one definitation, and two propositions previously established in (Agarwal et al., 2021). We denote the value function at the initial state $s \in S$ under policy $\pi$ as $V(s, \pi)$.

**Definition 2.** *The state visitation distribution $d_\mu^\pi(s)$ of a policy $\pi$ is defined as:*

$$d_\mu^\pi(s) = (1 - \gamma) \sum_{k=0}^{\infty} \gamma^k \mathrm{Pr}^\pi(S^{(k)} = s | S^{(0)} \sim \mu),$$

*where $\mathrm{Pr}^\pi(S^{(k)} = s | S^{(0)} \sim \mu)$ is the probability of visiting state $s$ at the $k$-th time step when the agent follows policy $\pi$ with an initial state distribution $\mu$.*

**Definition 3.** *The advantage function given policy $\pi$ $A : S \times A \to \mathbf{R}$ is defined as*

$$A(s, a, \pi) = Q(s, a, \pi) - V(s, \pi).$$

The value function, Q-value function, and advantage function, given reward $R(x)$ with side-payment $x$ and a parameterized policy $\pi_\theta$, is denoted by $V(s, \pi_\theta; x)$, $Q(s, a, \pi_\theta; x)$ and $A(s, a, \pi_\theta; x)$ respectively.

**Proposition 1.** *For direct parameterization, the partial derivative of a value function w.r.t. $\theta_{s,a}$ has an explicit form:*

$$\frac{\partial V(\mu, \pi_\theta; x)}{\partial \theta_{s,a}} = \frac{1}{\tau(1-\gamma)} d_\mu^{\pi_\theta}(s) Q(s, a, \pi_\theta; x). \tag{14}$$

*For softmax parameterization, we have*

$$\frac{\partial V(\mu, \pi_\theta; x)}{\partial \theta_{s,a}} = \frac{1}{1-\gamma} d_\mu^{\pi_\theta}(s) \pi(a|s) A(s, a, \pi_\theta; x), \tag{15}$$

*and*

$$\frac{\partial V(\mu, \pi_\theta; x)}{\partial \theta_{s,a}} = \frac{\pi_\theta(a|s)}{\tau} \frac{\partial V(\mu, \pi_\theta; x)}{\partial \pi_\theta(a|s)}. \tag{16}$$

**Proposition 2.** *The value function is L-smooth w.r.t. $\theta$. Under direct parameterization, for any $(\theta, \theta')$, and all starting state $s$,*

$$\|\nabla_\theta V(s, \pi_\theta; x) - \nabla_\theta V(s, \pi_{\theta'}; x)\| \leq \frac{2\gamma|A|}{(1-\gamma)^3} \|\theta - \theta'\|.$$

*Under softmax parameterization, for any $(\theta, \theta')$, and all starting state $s$,*

$$\|\nabla_\theta V(s, \pi_\theta; x) - \nabla_\theta V(s, \pi_{\theta'}; x)\| \leq \frac{8}{(1-\gamma)^3} \|\theta - \theta'\|.$$

After a straightforward derivation, we extend the above statement: for any initial distribution $\mu$, $\nabla_\theta V(\mu, \pi_\theta; x)$ is L-smooth under both direct and softmax parameterization.

## C    Proof of Corollary 2

To simplify the proof, we first state the following lemma, whose proof is straightforward and therefore omitted.

**Lemma 2.** *Assume a vector valued function with vector input $\varphi(p) = [\varphi_1(p), \ldots, \varphi_n(p)]^\mathsf{T}$. If $\varphi_1, \ldots, \varphi_2$ are all Lipschitz continuous w.r.t. the input variable $p$, then $\varphi$ is Lipschitz continuous w.r.t. the input variable $p$.*

*If a vector valued function with vector inputs $\varphi(p_1, \ldots, p_n)$ is Lipschitz continuous w.r.t. each $p_i$, $i \in \{1, \ldots, n\}$, then $\varphi(p_1, \ldots, p_n)$ is Lipschitz continuous w.r.t. $(p_1, \ldots, p_n)$.*

*The summation of Lipschitz continuous function is also Lipschitz continuous.*

By Lemma 2, we can simplify the proof by showing that $\nabla_\theta f$, $\nabla_x f$, $\nabla_\theta g$, and $\nabla_x g$ are Lipschitz continuous w.r.t. both $\theta$ and $x$.

To proceed, we state the following two lemmas as preparation, and the proofs are provided in Appendix D and E respectively.

**Lemma 3.** *The gradient of leader's value function $\nabla_\theta V(\mu, \pi_\theta)$ is Lipschitz continuous w.r.t. $\theta$.*

Recall that we use $\mathbf{d}_\mu^\pi$ to denote the vector with the entries to be $d_\mu^\pi(s, a)$ for each $s \in S$ and $a \in A$, given the policy $\pi$ and initial distribution $\mu$.

**Lemma 4.** $\mathbf{d}_\mu^{\pi_\theta}$ *is Lipschitz continuous w.r.t. $\theta$.*

*Proof.* The Lipschitz continuity that we can establish with the previous results is as follows:

- For $\nabla_x f$ w.r.t. $x$: This follows from Assumption 2.

- For $\nabla_\theta f$ w.r.t. $\theta$: This follows from Lemma 3.

- For $\nabla_x g$ and $\nabla_\theta g$: By Lemma 2, it suffices to prove the Lipschitz continuity of $\nabla_x V(\mu, \pi_\theta; x)$ and $\nabla_\theta V(\mu, \pi_\theta; x)$, where:
  - $\nabla_x V(\mu, x, \pi_\theta)$ is Lipschitz continuous w.r.t. $\theta$ based on Lemma 4, and independent of $x$.
  - The Lipschitz continuity of $\nabla_\theta V(\mu, x, \pi_\theta)$ w.r.t. $\theta$ is proven in Proposition 2 in Appendix B.

Therefore, we need only prove the Lipschitz continuity of $\nabla_\theta V(\mu, x, \pi_\theta)$ w.r.t. $x$. Then let's focus our analysis on $\dfrac{\partial V(\mu, \pi_\theta; x)}{\partial \theta_{s,a}}$. According to Proposition 1 in Appendix B, this partial derivative is linear in $Q(s, a, x, \pi_\theta)$ w.r.t. $x$ so that we only need to examine the properties of $Q(s, a, x, \pi_\theta)$.

We can express the $Q$-function as follows:

$$Q(s, a, x, \pi_\theta) = R(s, a; x) + \sum_{s' \in S} P(s'|s, a) V(s', \pi_\theta; x).$$

Since $R(s, a; x)$ is Lipschitz continuous w.r.t. $x$, we only need to examine the Lipschitz continuity for $V(s', \pi_\theta; x)$ w.r.t. $x$, which is implied by Lemma 1 given that the vector $\boldsymbol{d}_\mu^{\pi_\theta}$ has a bounded norm for any initial distribution $\mu$ and policy $\pi_\theta$. $\qquad\square$

## D   Proof of Lemma 3

To prove this lemma, we first restate Lemma 53 from (Agarwal et al., 2021) as Proposition 3.

**Proposition 3.** *(Smoothness of policy gradient) Consider a unit vector $u$, let $\pi_\alpha \triangleq \pi_{\theta+\alpha u}$ and let $\tilde{V}(\alpha)$ be the corresponding value at a fixed state $s_0$, i.e.*

$$\tilde{V}(\alpha) \triangleq V^{\pi_\alpha}(s_0).$$

*Assume that*

$$\sum_{a \in A} \left| \frac{d\pi_\alpha(a|s_0)}{d\alpha}|_{\alpha=0} \right| \le C_1, \quad \sum_{a \in A} \left| \frac{d^2 \pi_\alpha(a|s_0)}{(d\alpha)^2}|_{\alpha=0} \right| \le C_2,$$

*then*

$$\max_{\|u\|_2=1} \left| \frac{d^2 \tilde{V}(\alpha)}{(d\alpha)^2}|_{\alpha=0} \right| \le \frac{C_2}{(1-\gamma)^2} + \frac{2\gamma C_1^2}{(1-\gamma)^3}.$$

Next we provide a proof for Lemma 3.

*Proof.* Based on Proposition 3, we only need to show $\sum_{\boldsymbol{a} \in \boldsymbol{A}} \left| \frac{d\pi_\alpha(\boldsymbol{a}|s_0)}{d\alpha}|_{\alpha=0} \right|$ and $\sum_{\boldsymbol{a} \in \boldsymbol{A}} \left| \frac{d^2 \pi_\alpha(\boldsymbol{a}|s_0)}{(d\alpha)^2}|_{\alpha=0} \right|$ are bounded. We follow the idea in the proof of Lemma 54 and 55 in (Agarwal et al., 2021).

Let $h(\phi)$ be defined as $\prod_{j \in \mathbf{N} \setminus \phi} \pi_{\theta_j}(a_j|s_j)$, where $\phi$ is a subset of $\mathbf{N}$. It is clear that $0 \le h(\phi) \le 1$ for any $\phi \subseteq \mathbf{N}$. In the case of direct parameterization, let $n = |\mathbf{N}|$ is the total number of followers, and recall that $\mathbf{a} = (a_1, \cdots, a_n)$, we have

$$
\begin{aligned}
\sum_{\boldsymbol{a} \in \boldsymbol{A}} \left| \frac{d\boldsymbol{\pi}_\alpha(\boldsymbol{a}|\boldsymbol{s}_0)}{d\alpha}|_{\alpha=0} \right| &= \sum_{\boldsymbol{a} \in \boldsymbol{A}} \left| \sum_{j \in \mathbf{N}} \frac{\partial \boldsymbol{\pi}_\theta(\boldsymbol{a}|\boldsymbol{s})}{\partial \theta_{j,s_j,a_j}} \frac{d\theta_{j,s_j,a_j}}{d\alpha} \right| \\
&\overset{(i)}{\le} \sum_{\boldsymbol{a} \in \boldsymbol{A}} \sum_{j \in \mathbf{N}} \left| \frac{\partial \boldsymbol{\pi}_\theta(\boldsymbol{a}|\boldsymbol{s})}{\partial \theta_{j,s_j,a_j}} \right| \left| \frac{d\theta_{j,s_j,a_j}}{d\alpha} \right| \\
&\overset{(ii)}{\le} \sum_{\boldsymbol{a} \in \boldsymbol{A}} \sum_{j \in \mathbf{N}} |h(\{j\})| \\
&\le n|A|.
\end{aligned}
$$

where $(i)$ is follows from the triangle inequality and Cauchy–Schwarz inequality, and $(ii)$ is because $\frac{d\theta}{d\alpha} = u$ is a unit vector.

Also, differentiating again w.r.t. $\alpha$ gives

$$
\sum_{\boldsymbol{a}\in\boldsymbol{A}}\left|\frac{d^2\boldsymbol{\pi}_\alpha(\boldsymbol{a}|\boldsymbol{s}_0)}{(d\alpha)^2}|_{\alpha=0}\right| = \sum_{\boldsymbol{a}\in\boldsymbol{A}}\left|\sum_{j,k\in\mathbf{N}}\frac{\partial\boldsymbol{\pi}_\theta(\boldsymbol{a}|\boldsymbol{s})}{\partial\theta_{j,s_j,a_j}\partial\theta_{k,s_k,a_k}}\frac{d\theta_{j,s_j,a_j}}{d\alpha}\frac{d\theta_{k,s_k,a_k}}{d\alpha}\right|
$$

$$
\leq \sum_{\boldsymbol{a}\in\boldsymbol{A}}\sum_{j\neq k}|h(\{j,k\})|
$$

$$
\leq n(n-1)|A|.
$$

Let $\theta_s \in \mathbf{R}^{|A|}$ denote the parameters associated with a given state $s$ for a follower. In the case of softmax parameterization, from equation 10, we have $\nabla_{\theta_s}\pi_\theta(a|s) = \pi_\theta(a|s)(e_a - \pi(\cdot|s))$, where $e_a$ is an indicator vector with a 1 at the entry corresponding to action $a$ and $\pi(\cdot|s)$ is a vector of probabilities.

Then, we have

$$
\sum_{\boldsymbol{a}\in\boldsymbol{A}}\left|\frac{d\boldsymbol{\pi}_\alpha(\boldsymbol{a}|\boldsymbol{s}_0)}{d\alpha}|_{\alpha=0}\right| = \sum_{\boldsymbol{a}\in\boldsymbol{A}}|u^\mathsf{T}\nabla_\theta\boldsymbol{\pi}_\alpha(\boldsymbol{a}|\boldsymbol{s}_0)|_{\alpha=0}|
$$

$$
= \sum_{\boldsymbol{a}\in\boldsymbol{A}}\left|\sum_{i\in\mathbf{N}}h(\{i\})u^\mathsf{T}_{s_{i,0}}\nabla_{\theta_{i,s_0}}\pi_{\theta_i}(a_i|s_{i,0})\right|
$$

$$
\overset{(i)}{\leq} \sum_{\boldsymbol{a}\in\boldsymbol{A}}\sum_{i\in\mathbf{N}}h(\{i\})\left|\nabla_{\theta_{i,s_0}}\pi_{\theta_i}(a_i|s_{i,0})\right|
$$

$$
\leq \sum_{\boldsymbol{a}\in\boldsymbol{A}}\sum_{i\in\mathbf{N}}h(\{i\})\left|\pi_{\theta_i}(a_i|s_{i,0})(e_{a_i} - \pi(\cdot|s_{i,0}))\right|
$$

$$
\leq \sum_{i\in\mathbf{N}}\sum_{\boldsymbol{a}\in\boldsymbol{A}}\boldsymbol{\pi}_\theta(\boldsymbol{a}|\boldsymbol{s})\left|e_{a_i} - \pi(\cdot|s_{i,0})\right|
$$

$$
\leq \sum_{i\in\mathbf{N}}\max_{\boldsymbol{a}\in\boldsymbol{A}}\left|e_{a_i} - \pi(\cdot|s_{i,0})\right|
$$

$$
\leq 2n,
$$

where $(i)$ is follows from the triangle inequality and Cauchy–Schwarz inequality.

Also, for the softmax parameterization, the following first derivative and second derivative are bounded.

$$
\left|\frac{\partial\pi_{\theta_j}(a_j|s_j)}{\partial\theta_{j,s_j,\tilde{a}_j}}\right| \leq \frac{1}{\tau},
$$

and

$$
\left|\frac{\partial\pi_{\theta_j}(a_j|s_j)}{\partial^2\theta_{j,s_j,\tilde{a}_j}}\right| = \frac{1}{\tau}|\pi(\tilde{a}_j|s_j)(\mathbf{1}_{\tilde{a}=a} - \pi(\tilde{a}_j|s_j))(\mathbf{1}_{\tilde{a}=a} - 2\pi(\tilde{a}_j|s_j))| \leq \frac{2}{\tau}.
$$

Then, we have

$$\sum_{\boldsymbol{a}\in\boldsymbol{A}}\left|\frac{d^2\boldsymbol{\pi}_\alpha(\boldsymbol{a}|\boldsymbol{s}_0)}{(d\alpha)^2}|_{\alpha=0}\right| = \sum_{\boldsymbol{a}\in\boldsymbol{A}}\left|\sum_{j,k}\frac{\partial\boldsymbol{\pi}_\theta(\boldsymbol{a}|\boldsymbol{s})}{\partial\theta_{j,s_j,\tilde{a}_j}\partial\theta_{k,s_k,a_k}}\frac{d\theta_{j,s_j,\tilde{a}_j}}{d\alpha}\frac{d\theta_{k,s_k,a_k}}{d\alpha}\right|$$

$$\leq \sum_{\boldsymbol{a}\in\boldsymbol{A}}\sum_{j\neq k}\left|h(\{j,k\})\frac{\partial\pi_{\theta_j}(a_j|s_j)}{\partial\theta_{j,s_j,\tilde{a}_j}}\frac{\partial\pi_{\theta_k}(a_k|s_k)}{\partial\theta_{k,s_k,\tilde{a}_k}}\right| + \sum_{j=k}\left|h(\{j\})\frac{\partial\pi_{\theta_j}(a_j|s_j)}{\partial^2\theta_{j,s_j,\tilde{a}_j}}\right|$$

$$\leq \sum_{\boldsymbol{a}\in\boldsymbol{A}}\left(\frac{n|A|(n|A|-1)}{\tau^2} + \frac{2n|A|}{\tau}\right)$$

$$= \frac{n|A|^2(n|A|+2\tau-1)}{\tau^2}.$$

$\square$

## E  Proof of Lemma 4

*Proof.* To prove this lemma, we first present the following result and its proof:

**Lemma 5.** *With assumption 3, the value function $V(\mu,\pi_\theta;x)$ in a single-agent MDP is Lipschitz continuous w.r.t. $\theta$.*

From Assumption 3, the Q-value function is known to be bounded, and we denote its maximum value by $Q_{\max}$, Furthermore, Proposition 1 in Appendix B implies that the partial derivative of value function is bounded by $Q_{\max}/(1-\gamma)$ under direct parameterization, and $Q_{\max}/(\tau(1-\gamma))$ under softmax parameterization, where $Q_{\max}$ is the upper bound of Q-value function. This implies $\|\nabla_\theta V(\mu,\pi_\theta;x)\|$ is also bounded. Therefore, the value function $V(\mu,\pi_\theta;x)$ is Lipschitz continuous w.r.t. $\theta$.

We now proceed to prove the Lemma 4. We have

$$d_\mu^{\pi_\theta}(s,a) = (1-\gamma)\sum_{k=0}^\infty \gamma^k \mathrm{Pr}^\pi(S^{(k)}=s, A^{(k)}=a|S^{(0)}\sim\mu)$$

$$= \mathbb{E}_\pi\left[\sum_{k=0}^\infty \gamma^k(1-\gamma)\mathbf{1}(S^{(k)}=s, A^{(k)}=a) \mid S^{(0)}\sim\mu\right],$$

where $\mathbf{1}(\cdot)$ is the indicator function. Thus, for any $s\in S$ and $a\in A$, $d_\mu^\pi(s,a)$ can be viewed as a value function of policy $\pi$ evaluated with the following reward function

$$r(s',a') = \begin{cases}(1-\gamma), & \text{if } (s',a')=(s,a) \\ 0, & \text{otherwise}\end{cases}$$

From Lemma 5 (we choose $\|x\|=\mathbf{0}$), we derive that $d_\mu^{\pi_\theta}(s,a)$ is Lipschitz continuous w.r.t. $\theta$.  $\square$

## F  Proof of Corollary 3

Since $\nabla_\theta g(x,\theta) = [\nabla_{\theta_1}V_1(\mu,\pi_{\theta_1};x),\cdots,V_n(\mu,\pi_{\theta_n};x)]$, it suffices to show that each follower's value function $V(\mu,\pi_\theta;x)$ satisfies the PL-inequality w.r.t. $\theta$. i.e.

$$\|\nabla_\theta V(\mu,\pi_\theta;x)\|^2 \geq \kappa(V(\mu,\pi_{\theta^*};x) - V(\mu,\pi_\theta;x)),$$

where $\theta^* \in \arg\max_\theta V(\mu,\pi_\theta;x)$. We assume that the reward function is non-trivial (i.e., not identically zero). For any $\theta\neq\theta^*$, we can define a positive constant $\beta \triangleq \max_{(s,a)\in S\times A}\|\frac{\partial V(\mu,\pi_\theta;x)}{\partial\theta_{s,a}}\|$. It follows that that $\beta \leq \|\nabla_\theta V(\mu,\pi_\theta;x)\|$ because the norm of a vector is always greater than or equal to the absolute value of

any individual entry. From Proposition 1 in Appendix B, $\beta = \frac{1}{1-\gamma} \max_{(s,a)} d_\mu^{\pi_\theta}(s) \|Q^{\pi_\theta}(s,a)\|$ under direct parameterization, and $\beta = \frac{1}{\tau(1-\gamma)} \max_{(s,a)} d_\mu^{\pi_\theta}(s) \pi_\theta(a|s) \|A^{\pi_\theta}(s,a)\|$ under softmax parameterization. Thus, $\beta > 0$ under reasonable reward design.

By the performance difference lemma(Agarwal et al., 2021), we can show that the difference between value function evaluated at any policy $\pi_\theta$ and optimal policy $\pi_{\theta^*}$ is upper bounded.

$$
\begin{aligned}
V(\mu, \pi_{\theta^*}; x) - V(\mu, \pi_\theta; x) =& \frac{1}{1-\gamma} \sum_{s,a} d_\mu^{\pi_\theta}(s) \pi_\theta(a|s) A^{\pi_{\theta^*}}(s,a) \\
\leq& \frac{1}{1-\gamma} \sum_{s,a} d^{\pi_\theta^*}(s) \max_{\bar{a}} A^{\pi_\theta}(s,\bar{a}) \\
\overset{(i)}{\leq}& M,
\end{aligned}
\tag{17}
$$

where the upper bound $M$ is a positive constant and $(i)$ can be derived from the fact that both the state visitation frequency and advantage function are bounded quantities.

Thus, let $0 < \kappa \leq \frac{\beta^2}{M}$, for any $(x, \theta)$, we have

$$
\kappa(V(\mu, \pi_{\theta^*}; x) - V(\mu, \pi_\theta; x)) \leq \frac{\beta^2}{M} \cdot M = \beta^2 \leq \|\nabla_\theta V(\mu, \pi_\theta; x)\|^2.
$$

