# OpenReview forum: "Incentive Design for Multi-Agent Systems: A Bilevel Optimization Framework for Coordinating Independent Agents and Convergence Analysis"
_TMLR — Rejected by TMLR_

### Review · Reviewer_aRH4 · 2026-01-02

**Summary Of Contributions:**

This paper proposes a bilevel optimisation approach for incentive design in leader-follower games where the leader has a side-payments channel, and in which multiple independent followers are present. The paper focuses on a tabular setting, with a finite state and action spaces, and with tabular representations for the side-payments function, and shows the proposed bilevel optimisation approach converges to a stationary point under some reasonable assumptions. The functioning of the approach is demonstrated in a gridworld experiment.

## Strengths:
- Clearly written
- Method appears to be correct
- Proof appears to be correct

## Weaknesses:
- The method seems like it would become computationally expensive quickly for realistic problems.
- While the experiment does show that the approach does converge to a stationary point in a reasonable timeframe in practice, it could either be more realistic (providing evidence that the algorithm is practical) or more insightful (e.g. showing how the algorithm behaves in a wider range of tasks and comparing)
-  The difference with cited related work (e.g. Stadie et al 2020; Chen et al 2022) could be clearer

**Audience:**

Yes

**Audience Explanation:**

The problem tackled by this paper is relevant to various multi-agent systems settings. The authors cite a range of examples of the addressed problem occurring in practical situations.

The proposed approach does not seem trivial to me, though I do not have wide knowledge of literature on the specific problem tackled in this paper. The assumptions made are reasonable for some practical applications, though likely further works are needed to expand the applicability.

**Claims And Evidence:**

Yes

**Claims Explanation:**

I believe that the theoretical claims made in this paper are correct, and that the properties ascribed to the proposed approach are valid under the stated assumptions.

The experimental results do validate that the approach works in practice. However:
- The setting seems simplistic. The states at which the side-payments can be applied are hand-picked by the authors. Possibly this hand-picking is necessary for computational reasons (which should be discussed). Questions remain about how robust the algorithm is across different tasks, and how well the algorithm scales as the complexity of the task changes.
- Some important aspects of the experiments were unclear (see requested changes)

**Requested Changes:**

# Major (critical)
- Please add some discussion of the computation requirements of the method, e.g. a complexity analysis or discussion about the circumstances under which you expect the approach to be computationally feasible. For example, computing $\nabla_{x}q(x, \theta)$ seems like it may be very expensive in practice — if I've understood it correctly, we'd need to compute an optimal policy for each possible state-action pair for each follower.
- Please clarify the differences between initialisation 1, 2, and 3 of the experiment.
- Please add more discussion about why the different initialisation converge to such different values. If it is simply due to multiple stationary points, then perhaps also worth a discussion about the likelihood of reaching a global maximum, since the leader objective reported in Table 1 is very different across the initialisations. If you were to generate more initialisations of the experiment (however these are generated), would the algorithm keep converging to new stationary points?
- Please clarify the practical implications of non-unique optimal follower policies, and how your approach compares with those discussed in section 1.1 para 1, which ignore the possibility of non-unique optimal (or tie-break arbitrarily). Is it actually conceptually possible to use the approaches cited in that paragraph in your setting? If so, how do they compare in practice?

# Minor (strengthening)
- Fix minor typos (e.g. 'horizen' instead of 'horizon')

# To Consider (pure opinion)
- Consider using $V_l$ instead of $V$ to make it clearer when referring to the leader's value functions
- Sometimes indices were confusingly overloaded. E.g. in section 2.2 $\theta_s$ is a vector of $\theta$ for state $s$ (so $|A|$ elements), but in the same section $\theta_i$ is a vector of $\theta$ for all states and actions for follower $i$ (so has $|A|\times|S|$ elements). It may be worth considering choosing a different indexing scheme, or providing more clarifications.

---

> ### Author Response · Authors · 2026-02-25
>
> We thank the reviewer for the insightful comments and suggestions. We address the reviewer’s comments and requested changes in a point-by-point manner.
>
> **Weakness 1**
> We include the following analysis of per-step computational complexity of Algorithm 1 in the revised manuscript. Since the algorithm is a gradient-descent method, it is difficult to provide a closed-form analysis of the overall computational complexity
>
> `` Under direct parameterization, computing the policy gradient $\nabla_\theta V_l(\pi_\theta)$ has complexity $\mathcal{O}(m,|\tau|,n)$. Under softmax parameterization, this computation incurs complexity $\mathcal{O}(m,|\tau|,n|\mathcal{A}|)$, which is the same order as the complexity of computing $\nabla_\theta q(x,\theta)$.  The computation of the gradient $\nabla_x q(x,\theta)$ requires solving two occupancy-measure–related linear systems and has complexity $\mathcal{O}(2n|\mathcal{S}|^3)$.
> In addition, computing the optimal policy via value iteration for $n$ independent agents has complexity $\mathcal{O}(n|\mathcal{S}|^2|\mathcal{A}|)$. It is noted that the occupancy measure computation and value iteration are performed only once per incentive update. The independence of agents allows computations to be fully parallelized across agents, further improving scalability in practice.
>  Specifically, the dominant online computation—policy gradient evaluation—scales linearly with the number of agents and sampled time steps, i.e.,
> $\mathcal{O}(m|\tau|n)$ for direct parameterization and $\mathcal{O}(m|\tau|n|\mathcal{A}|)$ for softmax parameterization. As shown in the complexity analysis, all components of Algorithm 1 admit polynomial-time complexity. ''
>
> Based on the complexity analysis above, the proposed approach is expected to be computationally feasible under circumstances where the number of agents $n$, the state space $|\mathcal{S}|$, and the action space $|\mathcal{A}|$ are moderate.
>
> **Weakness 2**
> In the revised manuscript, we will include additional experimental results conducted in a Four Rooms environment~\cite{thoma2024contextual} with five agents in a $11\times11$ gridworld. In this setting, each agent $i$ has an individual goal state $G_i$, while the grey region represents a shared resting and charging area. The agents’ original reward structure is state-based: they receive a negative reward upon entering fire states, a relatively low positive reward in the charging area, and a higher positive reward when reaching their respective goal states.
>
> The leader’s objective is to mitigate traffic congestion at every time step. Concretely, the leader’s reward is negatively proportional to the number of agent pairs that are in close proximity to one another. Through an incentive mechanism implemented via side payments, the leader encourages agents to reduce congestion while explicitly accounting for the cost of providing incentives.
>
> If the reviewer considers this setting appropriate, we will include the corresponding experimental results and analysis in the revision to further demonstrate the scalability and robustness of our approach.
>
> **Weakness 3**
> In the related work, as mentioned in the paper, (Ji et al., 2023), (Stadie et al., 2020), and (Chen et al., 2022) assume a unique optimal solution for the lower-level problem and focus on a single-leader–single-follower interaction. However, it is well-known that the optimal policy in an MDP may not be unique. In contrast, we do not assume a unique solution for the lower-level problem and instead consider how to design incentive policies even when multiple optimal policies exist for the followers’ MDPs. This approach enables us to identify better incentive policies within a broader set of constraints.
>
> Due to the word limit, the remaining responses are provided in the next comment post.

---

> > ### Author Response · Authors · 2026-02-25
> >
> > **The setting seems simplistic. The states at which the side-payments can be applied are hand-picked by the authors. Possibly this hand-picking is necessary for computational reasons (which should be discussed). Questions remain about how robust the algorithm is across different tasks, and how well the algorithm scales as the complexity of the task changes.**
> > It is not necessary to manually select the locations of side payments. If no such selection is made, the dimension of the incentive vector is $|S| \times |A| \times N$, where $N$ is the number of followers. However, this full parameterization may lead to a large number of local optima for two reasons: (i) a given reward function may admit multiple optimal policies, and (ii) different reward functions can be policy-invariant, in the sense that they share the same set of optimal policies. As a result, the learned incentives may be difficult to interpret.
> >
> > In the experimental section, we therefore report only results that are interpretable among the experiments conducted. We could also include additional experiments in which the locations of side payments are selected randomly, and summarize the corresponding results in a table.
> >
> > **Requested change 1**
> > Please see the response in the “Weakness 1” section.
> >
> > **Requested change 2**
> > The difference between Initializations 1, 2, and 3 lies in how policy parameter $\theta$ is initialized, while the side payments vector is initialized identically in all cases. (1) The initial policy are set to the followers’ best response policies given initial side payments. (2) The initial policies are designed to maximize the leader’s value (Note that the optimal policy for our problem may not be the initial policies consdiering the cost of side payments); and (3) The initial policy for follower 2 is designed to have a low probability of reaching the less desired goal states, while the initial policy for follower 1 is configured to ensure a high probability of reaching the less desired goal states.
> >
> > We will clarify this in the revision.
> >
> > **Requested change 3**
> > We observe that the objective function exhibits many flat regions when using direct parameterization, compared to softmax parameterization. This helps to explain the different convergence trends under different initializations. The variability in outcomes primarily arises from the policy gradient method to estimating the gradient of the value function using sample trajectories. For initialization (2), the feasibility is satisfied in the first place, thus limited exploration causes the algorithm to converge to lower-cost regions within the flat region of the leader's objective function. In contrast, for other experiments, the initial policy violates the constraint, which encourages exploration and prompts $x$ to move beyond the current flat region. We plotted two surface plots illustrating that the relationship between $(x,\theta^\ast(x))$ (where $\theta^\ast(x)$ is the optimal joint policy that the leader would optimistically choose with $x$), and the value of the objective function for the leader. For simplicity, the dimension for $\theta^\ast(x)$ is omitted from the plot. The plot for direct parameterization exhibits many sharp jumps, whereas the softmax parameterization produces a much smoother function. From the surface plots, it appears that there are approximately four local optima.
> >
> > **Requested change 4**
> > If we assume that each follower has a unique optimal policy, there is no need to use a bi-level optimization approach; the problem then reduces to a single-level optimization that can be solved using standard gradient-based methods. However, the assumption of non-unique solutions is often more realistic in real-world applications. For example, in a ride-sharing platform, each passenger selects an optimal ride plan based on personal preferences, while the platform adjusts incentives for certain plans to encourage frequent usage or early bookings. In practice, a passenger may have multiple optimal ride plans, making the consideration of non-unique policies essential for designing effective incentive mechanisms.
> >
> > **To consider**
> > Thank you for the helpful suggestions. We will make the corresponding revisions to improve clarity.

---

> ### Author Response · Authors · 2026-03-10
>
> ***Updated response to weakness 2***
> We thank the reviewer for the helpful suggestion. We conducted an additional experiment in a Four Rooms environment [1] with five agents in an 11×11 stochastic gridworld, where agents may have different transition dynamics. The grey region represents a shared resting and charging area, while several fire states correspond to dangerous or undesirable regions (e.g., hazardous roads). The agents’ reward structure is state-based: they receive a negative reward when entering fire states, a small positive reward in the charging area, and a larger positive reward upon reaching the goal state.
>  Through an incentive design mechanism implemented via side payments, the leader encourages agents to reduce congestion while accounting for the cost of providing incentives. The leader aims to mitigate traffic congestion at each time step while also reducing the duration of congestion. Specifically, the leader’s reward is negatively proportional to the number of agent pairs that are in close proximity before all agents reach their goal states. As a result, the estimated leader value improves from −64.778 to −50.914, corresponding to a 21.402% improvement. We appreciate the reviewer’s suggestion and will include the details of this experiment in the revised version.
>
> [1]Thoma, V., Pásztor, B., Krause, A., Ramponi, G., & Hu, Y. (2024). Contextual bilevel reinforcement learning for incentive alignment. Advances in Neural Information Processing Systems, 37, 127369-127435.

---

### Review · Reviewer_FGT2 · 2026-01-10

**Summary Of Contributions:**

## Summary

This paper investigates incentive design in a multi-agent MDP setting with a single leader and multiple independent followers. The problem is formulated as a bilevel optimization, where followers individually optimize their own MDPs under side payments, and the leader optimizes a global objective that depends on the followers' best-response policies while accounting for incentive costs. To tackle challenges arising from non-concavity and multiple stationary points, the authors reformulate the problem as a constrained optimization and develop a first-order algorithm with theoretical convergence guarantees. Experimental results on a stochastic gridworld demonstrate convergence behavior and improvements in the leader's objective.

## Strengths

1. The paper is clearly written and well structured. The problem formulation, assumptions, and methodological contributions are presented in a logical and accessible manner.
2. The proposed algorithm is supported by rigorous theoretical analysis, including convergence guarantees to a stationary point of the original bilevel problem.

## Weaknesses

1. The assumption that followers are fully independent and do not interact with each other simplifies the problem setting and limits the applicability of the proposed framework to more general multi-agent systems with interactions.
2. The novelty of the algorithmic contribution is not entirely clear, especially given prior work on first-order methods for bilevel optimization (Liu et al., 2022). The paper would benefit from a clearer differentiation from existing approaches.

**Audience:**

Yes

**Audience Explanation:**

This paper is related to multi-agent RL, which is an important research topic.

**Claims And Evidence:**

Yes

**Claims Explanation:**

Theoretical and numerical results are provided to support the main claims of this paper.

**Requested Changes:**

1. At the beginning of Section 3.3, the authors present an example illustrating that Problem (6) can admit multiple stationary points. However, the role of this example is unclear, as it does not seem to directly inform the algorithm design or subsequent analysis. Clarifying its relevance would improve coherence.
2. The experimental evaluation is limited to a simple synthetic 6×6 gridworld. While this setting is useful for validating feasibility and convergence, it is insufficient to demonstrate the effectiveness of the method in more complex or realistic scenarios.
3. The paper lacks comparisons with existing methods, either theoretically or empirically. Without such comparisons, it is difficult to assess the practical advantages and relative contribution of the proposed approach.

---

> ### Author Response · Authors · 2026-02-25
>
> We thank the reviewer for the insightful comments and suggestions. We address the reviewer’s comments and requested changes in a point-by-point manner.
>
> **Weakness 1** We adopt the assumption of independent followers for two main reasons.
>
> First, this setting captures many real-world applications. For example, in ride-sharing platforms, each passenger selects an optimal travel plan based on individual preferences, while the platform adjusts incentives for certain plans to encourage frequent usage or early bookings. Another example arises in real-time pricing for smart grids with independent but heterogeneous users, where a price setter seeks to maximize social welfare subject to users’ independent best responses.
>
> Second, this assumption enables the derivation of an incentive design algorithm with convergence guarantees. If followers were to interact strategically, a common modeling approach would assume that they play a Nash equilibrium. However, in general non-cooperative games, Nash equilibria may be non-unique, and equilibrium selection can be misaligned across players. Moreover, to the best of our knowledge, existing multi-agent reinforcement learning methods do not guarantee convergence to a Nash equilibrium in general non-cooperative settings. Since equilibrium computation constitutes the lower-level problem in incentive design with interacting followers, these challenges significantly complicate both algorithm design and convergence analysis.
>
> **Weakness 2**
> The novelty of this paper lies in addressing the incentive design problem in multi-agent systems while providing convergence guarantees. In Liu et al. (2022), a solution was proposed for a general class of bilevel optimization problems with convergence guarantees; however, establishing these guarantees requires several assumptions on both the upper-level and lower-level objective functions. One of our main contributions is to show that, for the class of multi-agent incentive design problems considered in this work, these assumptions are satisfied when agents employ direct and softmax-parameterized policies. This result enables the application of bilevel optimization methods with provable convergence to a class of multi-agent systems with practical relevance.
>
> **Requested change 1**
> We aim to demonstrate that this problem has many local optima in preparation for our experiment. In the experiment, we demonstrate that our algorithm can converge to different local stationary points with different initializations.
>
> Due to the word limit, the remaining responses are provided in the next comment post.

---

> ### Author Response · Authors · 2026-02-25
>
> **Requested change 2**
> We include the following analysis of per-step computational complexity of Algorithm 1 in the revised manuscript. Since the algorithm is a gradient-descent method, it is difficult to provide a closed-form analysis of the overall computational complexity
>
> "Under direct parameterization, computing the policy gradient $\nabla_\theta V_l(\pi_\theta)$ has complexity $\mathcal{O}(m,|\tau|,n)$. Under softmax parameterization, this computation incurs complexity $\mathcal{O}(m,|\tau|,n|\mathcal{A}|)$, which is the same order as the complexity of computing $\nabla_\theta q(x,\theta)$.  The computation of the gradient $\nabla_x q(x,\theta)$ requires solving two occupancy-measure–related linear systems and has complexity $\mathcal{O}(2n|\mathcal{S}|^3)$.
> In addition, computing the optimal policy via value iteration for $n$ independent agents has complexity $\mathcal{O}(n|\mathcal{S}|^2|\mathcal{A}|)$. It is noted that the occupancy measure computation and value iteration are performed only once per incentive update. The independence of agents allows computations to be fully parallelized across agents, further improving scalability in practice.
>  Specifically, the dominant online computation—policy gradient evaluation—scales linearly with the number of agents and sampled time steps, i.e.,
> $\mathcal{O}(m|\tau|n)$ for direct parameterization and $\mathcal{O}(m|\tau|n|\mathcal{A}|)$ for softmax parameterization. As shown in the complexity analysis, all components of Algorithm 1 admit polynomial-time complexity. ''
>
> Based on the complexity analysis above, the proposed approach is expected to be computationally feasible under circumstances where the number of agents $n$, the state space $|\mathcal{S}|$, and the action space $|\mathcal{A}|$ are moderate.
>
> In the revised manuscript, we will include additional experimental results conducted in a Four Rooms environment~\cite{thoma2024contextual} with five agents in a $11\times11$ gridworld. In this setting, each agent $i$ has an individual goal state $G_i$, while the grey region represents a shared resting and charging area. The agents’ original reward structure is state-based: they receive a negative reward upon entering fire states, a relatively low positive reward in the charging area, and a higher positive reward when reaching their respective goal states.
>
> The leader’s objective is to mitigate traffic congestion at every time step. Concretely, the leader’s reward is negatively proportional to the number of agent pairs that are in close proximity to one another. Through an incentive mechanism implemented via side payments, the leader encourages agents to reduce congestion while explicitly accounting for the cost of providing incentives.
>
> If the reviewer considers this setting appropriate, we will include the corresponding experimental results and analysis in the revision to further demonstrate the scalability and robustness of our approach.
>
> **Requested change 3**
> To our knowledge, no existing work addresses bilevel incentive design in multi-agent system with non-unique followers' policies. We have carefully reviewed the literature and found no direct baselines that solve the same problem formulation. We would greatly appreciate the reviewer's suggestions on baseline comparison. If feasible, we will implement a suggested baseline and add to the revised version.

---

> ### Author Response · Authors · 2026-03-10
>
> ***Additional response to requested change 2***
> We thank the reviewer for the helpful suggestion. We conducted an additional experiment in a Four Rooms environment [1] with five agents in an 11×11 stochastic gridworld, where agents may have different transition dynamics. The grey region represents a shared resting and charging area, while several fire states correspond to dangerous or undesirable regions (e.g., hazardous roads). The agents’ reward structure is state-based: they receive a negative reward when entering fire states, a small positive reward in the charging area, and a larger positive reward upon reaching the goal state.
>  Through an incentive design mechanism implemented via side payments, the leader encourages agents to reduce congestion while accounting for the cost of providing incentives. The leader aims to mitigate traffic congestion at each time step while also reducing the duration of congestion. Specifically, the leader’s reward is negatively proportional to the number of agent pairs that are in close proximity before all agents reach their goal states. As a result, the estimated leader value improves from −64.778 to −50.914, corresponding to a 21.402% improvement. We appreciate the reviewer’s suggestion and will include the details of this experiment in the revised version.
>
> [1]Thoma, V., Pásztor, B., Krause, A., Ramponi, G., & Hu, Y. (2024). Contextual bilevel reinforcement learning for incentive alignment. Advances in Neural Information Processing Systems, 37, 127369-127435.

---

### Review · Reviewer_jWuu · 2026-02-11

**Summary Of Contributions:**

This paper addresses the problem of incentive design in multi-agent systems formulated as a bilevel optimization problem. The framework features a single leader (upper level) who provides side payments to influence the behavior of multiple independent followers (lower level), where each follower independently solves its own MDP.

The primary contributions of the paper include the problem formulation in a constrained optimization problem, a novel algorithm that relies on policy gradient methods, and a convergence analysis that uses PL arguments.

**Audience:**

Yes

**Audience Explanation:**

The topic of AI alignment through incentive design is of high interest to the TMLR community. The use of bilevel optimization to solve the Strong Stackelberg Equilibrium in MDPs is a relevant mathematical framework. However, the current "proof of concept" is too weak and the theoretical contribution is not strong enough.

Applying the model to complex domains like transportation networks or online platforms (as hinted in the intro) would significantly broaden the appeal and impact of the work.

**Broader Impact Concerns:**

I do not have any concerns about the ethical implications of the work.

**Claims And Evidence:**

No

**Claims Explanation:**

The paper builds heavily on the first-order approach by Liu et al. (2022). While the application to MAS with multiple MDP followers is a specific use case, the core algorithmic contribution lacks significant novelty over existing bilevel optimization techniques. Specifically, the authors should clarify how their approach differs fundamentally from previous multi-constraint bilevel algorithms like [1,2]. Since the followers' objectives are independent, the lower level effectively decomposes into a sum of value functions, making the comparison to existing constrained bilevel methods (like [1,2]) essential for establishing technical significance.

The current experimental evaluation is limited to a stochastic gridworld (6x6) with only two followers. This environment is too simple to demonstrate the scalability or robustness of the proposed method in the "real-world" scenarios mentioned in the introduction (e.g., ride-sharing platforms or smart grids). To support the claims, experiments should be extended to continuous state/action spaces and larger agent populations, to verify that the leader's objective remains computationally tractable as $n$ increases.

The presentation of empirical results is not good enough. The contour plots (Figure 2) and convergence graphs (Figures 4 & 5) have illegible axis labels, small fonts, and low resolution. The occupancy heatmaps (Figure 6) are particularly difficult to interpret due to poor image quality.

[1] Thoma, V., Pásztor, B., Krause, A., Ramponi, G., & Hu, Y. (2024). Contextual bilevel reinforcement learning for incentive alignment. Advances in Neural Information Processing Systems, 37, 127369-127435.

[2] Wu, Shuo, et al. "Robust reward design for Markov decision processes." Journal of Artificial Intelligence Research 84 (2025).

[3] Yang, Y., Gao, B., & Yuan, Y. X. (2025, April). Bilevel Reinforcement Learning via the Development of Hyper-gradient without Lower-Level Convexity. In International Conference on Artificial Intelligence and Statistics (pp. 4780-4788). PMLR.

**Requested Changes:**

I would like to propose the following changes:

1. Explicitly contrast the proposed algorithm with other state-of-the-art bilevel algorithms for non-convex lower levels. Why is this reformulation better?

2. Provide results on a more complex environment (e.g., a continuous control task or a larger-scale multi-agent coordination problem)

3. Replace all figures with high-resolution versions. Ensure that axis labels, legends, and titles are legible and that the color scales in heatmaps are clearly defined.

4. Include a study on how the regularization factor $w$ and the softmax temperature $\tau$  affect convergence speed and the quality of the leader's objective.

---

> ### Author Response · Authors · 2026-02-25
>
> We thank the reviewer for the insightful comments and suggestions. We address the reviewer’s comments and requested changes in a point-by-point manner.
>
> **Differences from previous work**
>
> Both papers [1] and [3] assume  that the lower-level problem admits a unique optimal solution.  Under this assumption, the bilevel problem can be reformulated as a single-level optimization problem, which can then be solved by computing the corresponding hypergradient.
> In contrast, our formulation allows for multiple optimal solutions at the lower level. In our setting, the leader adopts an optimistic stance: among all follower optimal responses, the leader assumes that the followers will select the one most favorable to the leader’s objective. Consequently, the resulting bi-level problem cannot be addressed by directly computing a standard hypergradient.
> Paper [2] considers a single-follower setting, whereas we extend the framework to multiple followers. Although each follower independently solves its own MDP, coordination emerges because the leader’s objective depends on the joint policy of all followers. This structural coupling prevents the reformulation used in [2]. In particular, the leader’s objective cannot be decomposed with respect to individual follower policies, and therefore the problem cannot be reduced to a linear program as in [2]. In addition, Paper [3] considers only a single follower.
>
> In summary, our problem formulation generalizes these papers, extending to the multi-solution (optimistic) bilevel setting and the multi-follower case.
>
> **Experimental Setup**
>
> In the revised manuscript, we will include additional experimental results conducted in a Four Rooms environment~\cite{thoma2024contextual} with five agents in a $11\times11$ gridworld. In this setting, each agent $i$ has an individual goal state $G_i$, while the grey region represents a shared resting and charging area. The agents’ original reward structure is state-based: they receive a negative reward upon entering fire states, a relatively low positive reward in the charging area, and a higher positive reward when reaching their respective goal states.
>
> The leader’s objective is to mitigate traffic congestion at every time step. Concretely, the leader’s reward is negatively proportional to the number of agent pairs that are in close proximity to one another. Through an incentive mechanism implemented via side payments, the leader encourages agents to reduce congestion while explicitly accounting for the cost of providing incentives.
>
> If the reviewer considers this setting appropriate, we will include the corresponding experimental results and analysis in the revision to further demonstrate the scalability and robustness of our approach.
>
> **Requested changes 1**
> We respectfully note that, to the best of our knowledge, there are currently no methods for incentive design in leader-multi-follower games that address lower-level problems with multiple optimal solutions under general nonconvex objectives and constraints.
>
> **Requested changes 2**
> Please see the response in the “Experimental Setup” section.
>
> **Requested changes 3**
> Thank you for the suggestion. In the revised manuscript, we will replace all figures with high-resolution versions and improve the readability of axis labels, legends, titles, and heatmap color scales to ensure clarity.
>
> **Requested changes 4**
> The weight $w$ represents the regularization factor for the cost of side payments. Intuitively, when $w$ is larger, the leader incurs a higher cost to incentivize the group of followers. As a result, the leader is less likely to provide many side payments, which may lead to worse overall performance.
>
> In finite MDPs, there always exists an optimal deterministic policy (often referred to as the hardmax policy), which selects actions that maximize the state-action value function. The softmax policy introduces a temperature parameter $\tau > 0$ that controls the degree of stochasticity in action selection. As $\tau \to 0$, the softmax policy increasingly concentrates probability mass on the maximizing actions and converges to the deterministic hardmax policy. Conversely, larger values of
> $\tau$ produce more  stochastic behavior.
>
> We will include results on the convergence rate and evaluate the performance of the agents according to the leader's value function, considering different values of the weight $w$ and the temperature parameter $\tau$.

---

> ### Author Response · Authors · 2026-03-10
>
> ***Requested changes 2***
> We thank the reviewer for the helpful suggestion. We conducted an additional experiment in a Four Rooms environment [1] with five agents in an 11×11 stochastic gridworld, where agents may have different transition dynamics. The grey region represents a shared resting and charging area, while several fire states correspond to dangerous or undesirable regions (e.g., hazardous roads). The agents’ reward structure is state-based: they receive a negative reward when entering fire states, a small positive reward in the charging area, and a larger positive reward upon reaching the goal state.
>  Through an incentive design mechanism implemented via side payments, the leader encourages agents to reduce congestion while accounting for the cost of providing incentives. The leader aims to mitigate traffic congestion at each time step while also reducing the duration of congestion. Specifically, the leader’s reward is negatively proportional to the number of agent pairs that are in close proximity before all agents reach their goal states. As a result, the estimated leader value improves from −64.778 to −50.914, corresponding to a 21.402% improvement. We appreciate the reviewer’s suggestion and will include the details of this experiment in the revised version.
>
> [1]Thoma, V., Pásztor, B., Krause, A., Ramponi, G., & Hu, Y. (2024). Contextual bilevel reinforcement learning for incentive alignment. Advances in Neural Information Processing Systems, 37, 127369-127435.

---

### Decision · Action_Editor_fm23 · 2026-06-07

**Recommendation:** Reject

**Audience:**

Yes

**Audience Explanation:**

Reviewers found the paper generally well written and believe it would be of interest to some members of the community as long as the claims & evidence issues are addressed.

**Claims And Evidence:**

No

**Claims Explanation:**

The reviewers have asked for additional experiments supporting the claim that the current method is an improvement over the state of the art (including algorithms that do not formally support multiple followers) to prove existing methods cannot be applied directly to their setting. While a description of an additional experiment is provided in the comments, this has no been presented in a revision of the submission. Similarly, the clarifying statement requested by reviewer aRH4 are not included in an updated revision. It then seems the manuscript requires a major revision in order to incorporate all the additional discussions in the comments and be suitable for acceptance.

**Resubmission Of Major Revision:**

The authors may consider submitting a major revision at a later time.